# Smartvessel: A New Extinguisher Prototype Based on New Materials and IoT Sensors

**DOI:** 10.3390/s23063134

**Published:** 2023-03-15

**Authors:** Javier Pisonero, Enrique González-González, Roberto García-Martín, Diego González-Aguilera

**Affiliations:** 1Department of Cartographic and Land Engineering, Higher Polytechnic School of Ávila, Universidad de Salamanca, 05003 Ávila, Spain; 2Department of Mechanical Engineering, Higher Polytechnic School of Zamora, Universidad de Salamanca, 49022 Zamora, Spain

**Keywords:** extinguishers, new materials, CFRP, AFRP, additive manufacturing, IoT sensors

## Abstract

Smartvessel is an innovative fire extinguisher prototype supported by new materials and IoT technology that seeks to improve the functionality and efficiency of conventional fire extinguishers. Storage containers for gases and liquids are essential for industrial activity as they enable higher energy density. The main contributions of this new prototype are (i) innovation in the use of new materials that provide lighter and more resistant extinguishers, both mechanically and against corrosion in aggressive environments. For this purpose, these characteristics are directly compared in vessels made of steel, aramid fiber and carbon fiber with the filament winding technique. (ii) The integration of sensors that allow its monitoring and provide the possibility of predictive maintenance. The prototype is tested and validated on a ship, where accessibility is complicated and critical. For this purpose, different data transmission parameters are defined, verifying that no data are lost. Finally, a noise study of these measurements is carried out to verify the quality of each data. Acceptable coverage values are achieved with very low read noise, on average less than 1%, and a weight reduction of 30% is obtained.

## 1. Introduction

The storage of liquids and gases in pressure vessels is an important part of industrial activity today. Storing at high pressures, a higher energy density is achieved and, therefore, better energy–volume ratios. This has an impact on industrial performance: (1) greater autonomy of the user devices; (2) the optimization of transport and logistics; (3) increased efficiency in energy displacement; and (4) a reduced carbon footprint and lower environmental impact.

However, current standard extinguisher solutions have several drawbacks [1]: (1) they are heavy to handle; (2) they are prone to corrosion; and (3) they have an inefficient volume-to-weight ratio, decreasing their economic and environmental viability.

Typically, extinguishers are manufactured using alloy steel due to the knowledge in terms of properties and manufacturing techniques. This results in very heavy containers, where approximately 30% of a useful extinguisher is the weight of the empty bottle. Therefore, new manufacturing techniques are being studied, as well as different types of materials with a better strength-to-weight ratio. Additive manufacturing has found here an opportunity with the filament winding technique with composite materials [2,3,4], such as carbon-fiber-reinforced polymers (CFRPs), glass-fiber-reinforced polymers (GFRPs) or aramid-fiber-reinforced polymers (AFRPs), which have high strength and a much lower weight than bottles made of steel alloys [5].

Another problem to which conventional extinguishers are exposed is corrosion. Many of these pieces of equipment are located outdoors, in areas with high humidity, and in chemical environments that directly attack the steel surface, even the stainless steel [6]. In this sense, composite materials have demonstrated a great capacity to resist corrosion, even in very aggressive environments such as marine areas [7,8]. In addition, both for their mechanical behavior and chemical resistance, parameters can be varied in manufacturing to find an optimum design for each environment.

On the other hand, smart maintenance approaches, called Maintenance 4.0 [9], are being developed in the context of the Industry 4.0 paradigm and are fed by the most advanced tools in the field of data processing and storage, adding operational information concerning the context and large data sets that can be processed with techniques of the Internet of Things (IoT), artificial intelligence and big data. However, this predictive and intelligent maintenance approach is still pending in the fire extinguisher industry [10].

Regardless of the material, in fire extinguishers, it is mandatory to check through revisions of all their mechanical elements and the state of charge of the extinguisher to ensure proper service conditions. These revisions, which are necessary for the extinguisher to fulfill its mission in case of fire or explosion, are carried out by a specialized operator and performed periodically. This time-consuming task could be avoided with an intelligent real-time monitoring system, which would also contribute to economic savings [11], especially in hazardous and/or inaccessible areas (e.g., wind turbine towers, ships, etc.), where the inspection costs are very high. Although there are different initiatives for monitoring the status of fire extinguishers [11,12,13], the approach developed by the authors is only focused on detecting pressure changes as a parameter to be monitored and always under conventional extinguishers. Furthermore, they use wireless communication networks, which are limited due to coverage in complex scenarios (e.g., ships). These communication problems can be solved with low-power wide-area networks (LPWANs), which are characterized by achieving long ranges in communication with low energy consumption and low data transfer speed. Therefore, it is not a technology designed to transmit audio or video signals, but it is ideal for providing connectivity to devices in remote locations, powered by a battery and with a low volume of data transmissions [14]. Within LPWAN networks, there are different types of technologies (e.g., LTE-M, LoRaWAN, Sigfox and NB-IoT) [15], which have already been tested in different industrial applications [16,17,18,19]. While technologies that use licensed frequency bands, such as NB-IoT or LTE-M, excel in transmission time and roughness, technologies that use unlicensed frequency bands (i.e., ISM bands), such as Sigfox or LoRaWAN, stand out for network capacity, lower cost and device lifetime [20]. Specifically, LoRaWAN networks are configured with a star topology, where a gateway receives messages from different devices or nodes. Transfer rates range from 10 to 100 Kbps, transmitted data frames are tens of bytes and the transmission cycle is limited to prevent saturating ISM unlicensed frequency bands [21].

In this work, different extinguishers manufactured with new materials and methods are tested and analyzed. In addition, we design and integrate different IoT sensors so intelligent monitoring can be applied. As a result, a final prototype is designed and validated for providing the automatic and remote maintenance of the extinguisher, making the inspection tasks more efficient, saving costs and contributing to the optimization of logistics, transport and sustainability.

Finally, a study of the characteristics and behaviors of the different new materials is carried out, which is later evaluated on a pressure bottle with similar characteristics in terms of operation. Furthermore, a study and development of the sensor system that best suits this case study and its possibility of being embedded into the system is performed.

In this work, we seek to provide an answer to the ability to provide an IoT structure in complex environments, such as a large vessel. This paper is structured as follows: After this Introduction, from which the main contribution of the manuscript is motivated, Section 2 describes, in detail, the new materials and methods developed; Section 3 outlines and discusses the main results achieved. The final section is devoted to highlighting the main conclusions and future perspectives of the approach.

## 2. Materials and Methods

### 2.1. New Materials for Fire Extinguishers

As was mentioned, fire extinguishers are mostly manufactured from steel alloys. For this purpose, steel sheets with a thickness of 1 to 1.5 mm were conformed. The basic shape of the vessel was achieved by joining the two halves of the vessel by shaping, as shown in Figure 1. The two halves were joined by MIG/MAG wire welding, and the upper thread was prepared for the drive valve. The vessel was finally painted with RAL 3000 epoxy paint with a minimum thickness of 110 microns to prevent corrosion. In general, the weight of the bottle is about one-third of the total mass, with the rest being the extinguishing agent.

To make it more resistant and lighter, equipment was manufactured with new composite materials, generally based on epoxy matrices, and with the filament winding manufacturing technique. This technique consists of winding on a mandrel of pre-impregnated fibers of this epoxy matrix. The results are products called composite overwrapped pressure vessels (COPVs). These vessels are classified into different types depending on the type of material and the inner vessel, called Liner. The different types of COPV are shown in Table 1.

Costs differ depending on the type of COPV used, with type V being the most expensive because more composite is needed for shaping and interior finishing. In all cases, the fibers support the full structural capacity of the vessel.

Depending on the type of application, more or less mechanical strength of the material to support the efforts is required. For the most common applications, such as wood or liquid acting as fuel, low-pressure extinguishers (i.e., 17 bar), mainly powder and hydric, are used; meanwhile, for fires where the fuel is gaseous, high-pressure extinguishers (i.e., 120 bar), mainly CO_2_ extinguishers, are used. With the aim of using new materials in applications of this type, it is necessary to use type IV COPVs that guarantee structural resistance and weight reduction. In addition, these materials must have a minimum plastic deformation capacity. Therefore, the fibers used for the manufacture of these vessels should be carbon, aramid and glass fibers (Table 2) [5]:

Composites need a matrix to distribute the stresses, and epoxy matrices are usually used for the selected fibers [22]. These materials have different mechanical properties; therefore, from the rule of mixtures [23], it is possible to calculate the mechanical characteristics of the composite material, as well as the different combinations to obtain the desired configuration. In addition, there are manufacturing parameters, such as the number of layers, fiber orientation [24], etc., that significantly vary the performance of the final material of the product.

Differentiating between low-pressure extinguishers (i.e., 17 bar) and high-pressure extinguishers (i.e., 120 bar), the dimensions and bottle designs for these different types of equipment vary considerably. The material for manufacturing the equipment will have a considerable effect on the total weight of the equipment and the maximum allowable pressure. Therefore, a comparison of the different materials of interest for this equipment is made in Table 3.

### 2.2. Accessories, Sensors and Connectivity

For a correct verification of the condition of the extinguishers, it is necessary to distinguish between the different types of equipment on the market. On the one hand, there are extinguishers whose extinguishing agent is powder or water, and the propellant is nitrogen in the gaseous phase. On the other hand, there are CO_2_ extinguishers whose extinguishing agent itself is the gas that displaces oxygen and interrupts the combustion process. In nitrogen extinguishers, the pressure value is constant, as has been established according to regulations around 17 bars [25], while in CO_2_ extinguishers, the pressure varies considerably. This is due to the fact that, inside the container, CO_2_ is in interphase, having fluid in liquid and gas phases at the same time, so the pressure value fluctuates constantly and is susceptible to change with the temperature of the environment. Therefore, the inspections of the different types of extinguishers are different: in those using nitrogen as a propellant, it is important to validate the value of the pressure, while in CO_2_ extinguishers, it is necessary to check the weight of all equipment to rule out leaks.

#### 2.2.1. Sensors

Different types of sensors have been used to measure the different parameters necessary to verify the status of the extinguishers. To measure pressure, three methods were used: (i) A commercial pressure sensor, in the 0–34 bar range, which is a low-cost solution; (ii) a strain gauge applied on the external part of the extinguisher. This is an expensive solution, but it requires less space and can works in a wider pressure range (i.e., 0–140 bar) (Figure 2a); (iii) an ad hoc hook and special base printed in 3D for wall-mounted extinguishers (<20 kg) and bigger extinguishers placed on the floor (<100 kg), respectively (Figure 2b). The strain gauge used was a 120 Ω strain gauge from Hottinger Baldwin Messtechnik (HBM) ^®^ (Dasrmstadt, Germany). (Figure 2c).

For measuring the internal pressure, the strain gauges were placed directly on the shell of the extinguisher. By applying the thin-walled rule in cylindrical vessels (Equation (1)), we can obtain the tensional state as a pressure function. Considering that the material works in an elastic regime, through Hooke’s law (Equation (1)), we can express the deformation of the fire extinguisher as a pressure function (Equation (2)). Therefore, by knowing the material behavior, the inner pressure can be known by directly measuring the deformations of the material (Equations (1) and (2)):(1)$$\sigma_{1}=\frac{Pr}{t};\;\sigma_{2}=\frac{Pr}{2t},\;where\;\sigma =E\cdot \varepsilon$$
(2)$$P=\frac{\sigma_{1}\cdot t}{r};P=\frac{E\cdot \varepsilon_{1}\cdot t}{r}$$
where:$\sigma_{1}\equiv$ transversal stress;$\sigma_{2}\equiv$ radial stress;$P_{r}\equiv$ pressure;t≡ thicknessE≡ Young’s modulus of the material;ε≡ strain.

All the sensors using strain gauges operate based on a Wheatstone bridge [26]. In this bridge, where the strain gauge is one of the resistors, a balance is obtained. (Figure 3).

Any variation in the resistances translates into a variation in the voltage reading at the ends of the circuit. Therefore, the gauge will vary this bridge, and the voltage variation across the circuit can be read. This requires an amplifier to serve as a translator of this variation. For this purpose, an HX711 AD amplifier was used for all cases.

#### 2.2.2. Accessories

As explained before, for extinguishers where nitrogen is the propellent agent, it is important to know the value of the pressure within the container. Meanwhile, for CO_2_ extinguishers, it is important to know the total weight of the equipment so that possible leaks or damage to the shell can be ruled out. For this reason, three accessories were designed and manufactured with 3D printing to carry out the verification process of the equipment that incorporates different sensors connected by LoRaWAN.

On the other hand, an accessory was designed to be attached directly to the hook of the extinguisher to measure its weight. It consists of a piece made of Polyethylene Terephthalate Glycol (PETG) [27] and was printed in 3D with an Ender© 3 pro printer (Creality 3D, Shenzhen, China) that had a nozzle diameter of 4mm, accepts 1.75mm filament and a print volume of 220 × 220 × 250 mm [28]. This piece can be hung from a standard hook that is anchored to a wall and serves as an interconnection with the equipment, being compatible with any extinguisher and commercial solution (Figure 4). The weight sensor was placed so that the weight of the extinguisher applies a force perpendicular to it and can correctly measure its weight. Thus, even if there are movements (e.g., extinguishers placed on ships), the measurement of this sensor is correct, avoiding large fluctuations. All the accessories were manufactured with the following parameters: 6 perimeters, 5 bottom layers and 5 top layers, a grid filler with a 15% filler and a layer height of 0.2 mm [29].

To measure the state of the CO_2_ extinguishers, a longitudinal load sensor was used based on strain gauge technology. A platform was designed (Figure 4) and printed in 3D, with PETG acting as a scale, where the weight was measured directly.

#### 2.2.3. Connectivity

Regarding communication networks, a LoRaWAN network was used. This network was chosen due to the great synergy between the characteristics of this network and the application requirements given in this work: long-range connectivity and energy autonomy, which implies low power consumption. Another reason for choosing LoRaWAN network was its scalability for being connected with other devices, thus, improving its coverage by adding more gateways [30].

An Arduino^®^ MKR WAN 1310 (Arduino, Turin, Italy) was used to develop the device and to obtain and send the data. This device was placed in a small housing printed in 3D with PETG so that it could be easily placed anywhere near the extinguisher to obtain the sensor readings. Inside this board (Figure 3), an HX711 AD amplifier was connected, working as an interface for weight and deformation measurements together with some connectors for the pressure sensors. As a result, a part of the board was dedicated to the connection with the weight sensors and another to the connection with the pressure sensors, being a compact and multi-reading system. Next to the board, a compact and manageable battery provided independence and autonomy.

The choice of this microcontroller was motivated due to the fact that the IoT connection can be realized with LoRaWAN protocol. This protocol fits perfectly to the need of this application since the data size was not large. The transfer distance range needs to be wide, and finally, the power consumption must be small since the board was supported by a battery, and this must allow the battery life to be prolonged over time.

Last but not least, a small box with an antenna was designed and manufactured in 3D printing, integrating the microcontroller and the amplifier board. This box can be placed near the extinguisher or anywhere nearby without any problem. Different slots were designed to allow the different sensors to be connected to the corresponding pins without any mistakes (Figure 5).

## 3. Results

### 3.1. New Materials for Fire Extinguishers

Different bottles were manufactured, analyzed and compared with traditional bottles dedicated to fire extinguishers.

#### 3.1.1. Steel Vessel

A steel bottle with a thickness of 1.5 mm was used as a test model. The characteristics of this bottle can be seen in Table 4:

This is a bottle dedicated to storing powder or water as an extinguishing agent, with a total load of 6 kg. This bottle withstands an internal pressure of about 70 bar.

#### 3.1.2. Aramid Vessel

A polymeric liner of polypropylene (PP) was used for the aramid COPV. An aramid fiber of 0.2 mm thickness was wound around it. This winding was made using the wet filament winding technique with the following configuration: A fiber angle of [90°/45°/–45°/45°/45°]. The epoxy resin, Sicomin© SR1500 (Sicomin, Chateauneuf les Martigues, France), was used as the matrix. The result was a bottle with the following characteristics (Table 5):

This is a bottle dedicated to storing powder or water as an extinguishing agent, with a total load of 6 kg. This bottle withstands an internal pressure of about 68 bar.

#### 3.1.3. Carbon Fiber Vessel

The commercial carbon fiber, Toray© CC 200T-120 (Toray Industries, Tokyo, Japan), was used with a thickness of 0.2 mm as reinforcement in a polymeric matrix made with a commercial epoxy resin, Sicomin© SR1500 (Sicomin, Chateauneuf les Martigues, France), and in combination with Hardener Biresin© SD2505 (Sika, Bar, Switzerland). The manufacturing parameters were a fiber angle of [−45/45°]_10_, a winding speed of 20 RPM and a curing temperature of 50°C. The result is a bottle with the following features (Table 6):

This is a bottle dedicated to storing powder or water as an extinguishing agent, with a total load of 6 kg. This bottle withstands an internal pressure of about 68 bar. The different vessels are shown in Figure 6.

In brief, the results of the comparison of these three materials are outlined in Table 7:

It should be noted that the added value of COPVs type IV (Table 1) is that they can store more pressure, so that, despite having less extinguishing agent, they can be used for a long time, providing similar benefits to more voluminous steel bottles, reducing its dimension and weight considerably.

### 3.2. Accessories, Sensors and Connectivity Results

As it was stated above, a distinction was made between accessories and sensors for powder/water extinguishers and CO_2_ extinguishers.

#### 3.2.1. Powder and Water Extinguishers

To check the condition of a foam or water extinguisher, it is necessary to know the internal pressure of the bottle. These extinguishers, in addition, are usually hung on a hook so as to know their weight or to know if the extinguisher is in the correct state or has been used and, therefore, if it is necessary to recharge or replace it.

For this purpose, pressure and load sensors were used, as explained above. The pressure sensor has a conical thread, which can be inserted directly into the pressure gauge slot of the valve. However, being tapered, it does not fit perfectly with the groove. For this reason, a connector was designed and manufactured to guarantee the tightness of a toroidal joint and a piece that serves as a union and adapter between threads.

For the load sensor, an ad hoc accessory was designed and manufactured, which serves as an interconnection between the wall hook and the extinguisher itself (Figure 7). In this hook, a load sensor was placed, which, through strain gauges, measures the deformation at this point and transcribes it directly to the applied load. This accessory is sensitive to the placement of the extinguisher, but by applying sufficient sensitivity and pressure, it is useful for checking if the extinguisher is loaded or if it has been removed from the hook.

On the other hand, strain gauges were installed directly on the vessel surface. In this way, by measuring the axial and transverse deformation, the internal pressure of the extinguisher can be determined. Specifically, the thin-walled theory can be applied.

#### 3.2.2. CO_2_ Extinguishers

CO_2_ extinguishers contain interphase fluid. The gas phase has great variability in terms of internal pressure, so measuring this parameter directly is not the most convenient way. For this reason, the total weight of the extinguisher must be measured directly. For this purpose, a device whose performance is similar to a scale was designed and manufactured with a load sensor inside so that the extinguisher can be placed on top and weighed directly. An ad hoc extinguisher charging system was developed that allows the internal charge of the equipment to be varied while it is being weighted by means of a pressure circuit.

#### 3.2.3. Connectivity

The validation of connectivity and coverage tests were performed on a dredge ship located in a shipyard in Pontevedra (Spain). The LoRaWAN network coverage was tested by the placement of two antennas: one in the bridge (inside the cabin) and the other one in the highest area of the ship over the bridge (outside). In order to carry out the tests, data were sent from different points on the ship (Figure 8) with different levels of power transmission signal or spread factor (SF 7, SF 9 and SF 11) with the aim of comparing the signal and noise levels received by the different gateways installed (Table 8, Table 9 and Table 10).

#### 3.2.4. Monitoring Results

To monitor and graphically represent the data in real time, a dashboard was developed, showing the different measurements through time. This dashboard is fully adaptable and, in this case, has been designed to read weight and pressure, separately. The pressure sensor has absolute reading; therefore, it is not necessary to calibrate it at any time, and when connected, it will directly give correct readings in bar. However, for the load sensors, a prior calibration is required. For this purpose, a tare protocol was designed, which consists of connecting the sensor without any load. The data acquisition is allowed to stabilize for one minute because noise appears when the sensor is connected. After one minute, a fire extinguisher filled with a known weight will be placed for one minute to avoid the same noise. Once the taring protocol is completed, the sensor is ready to monitor data. In this case, the following programs, all of them open source, will be used: NodeRed software to make a connection through the Mqtt protocol and transmit the data to an Influxdb database with JavaScript. Once the data are registered in the database, the data are visualized through the Grafana platform (Figure 9).

Regarding data sending, for prototyping and functionality verification, a sending frequency of 10 s was defined. However, so that it can be used over a long period of time with batteries connected, the most advisable frequency is an automatic sending of one data acquisition per day, and a button will be included to send the data instantaneously if necessary. The statistical values of every step are shown in Table 11.

## 4. Conclusions

The use of new materials in marine environments and even in mixed environments (e.g., oil tankers, transport of chemical compounds, etc.) is a viable alternative due to their resistance to corrosion, being much more cost-effective than the use of conventional materials (e.g., steel and stainless steel) because their useful life is much longer. On the other hand, there are additional advantages: These solutions are lighter, resulting in more ergonomic devices and lower logistics costs. Sensors can also be integrated more easily.

The use of the LoRaWAN network has proven to provide coverage at all locations on a ship, even in less accessible areas such as the engine room. The use of the LoRaWAN network allows for increased network capacity and coverage, as seen in field tests on a ship. Therefore, this solution is easily scalable to any situation by adapting the number of gateways to the required coverage area.

The use of low-cost technology has proven to be effective for the proposed application, where great accuracy is not required but we must guarantee a low-cost solution given the number of extinguishers to be monitored (e.g., 400 in the case of tankers). LoRaWan has the advantage of being a scalable system, which allows the incorporation of additional devices depending on the demand of the application. At the same time, these low-cost devices based on the use of strain gauges are very versatile because different parameters can be measured with the same platform without any additional interface. The devices used, in turn, meet the requirement of low energy consumption, which makes them very suitable for preventive maintenance applications. The developed device could incorporate new parameters, such as temperature, humidity or CO_2_ concentration, thereby adding more capacities for future possibilities, where these new signals treated from an artificial intelligence system, specifically in the case study presented in this work, allow beyond preventive maintenance to predict risk situations on ships (fires, gas leaks, leaks, structural failure, etc.).

The combination of the study of materials and wireless sensing allows a significant improvement in the manageability and predictive maintenance of the elements. The level of noise obtained by the sensors was sufficiently small to be able to obtain very good results, with SF7 being the one that gives the best results. Analyzing the signal and noise level data received by the gateways, there was no data loss for the gateway located on the bridge (outside); although, for the gateway located in the cabin (inside), there was a loss of some packets sent from the furthest positions (Table 8, Table 9 and Table 10).

On the other hand, the proposed system has some limitations: First, the sensitivity of the strain gauge requires a special circuit to amplify the signal, so we had to develop a special circuit to obtain the appropriate gain. In addition, it was necessary to protect the strain gauge from the environment. Another aspect to note is that each measurement will be made individually from each piece of equipment, using for each one a microcontroller. This is due to the use of amplifier boards that occupy a high number of pins in the microcontrollers. However, because this is a prototype, this aspect will be taken into account in future work in order to be able to make multiple measurements using as few controller boards as possible.

In this way, a much more complete predictive context can be created. In addition, numerical models can also be implemented through the finite element method (FEM). Thus, a mechanical behavior predictive model of the prototype (fire extinguisher) design will be available. Another future step will be managing all the collected data through artificial intelligence techniques to create a predictive algorithm that allows knowing the condition of the extinguishers and also to anticipate situations where the possibility of fire occurrence is high.

## Figures and Tables

**Figure 1 sensors-23-03134-f001:**
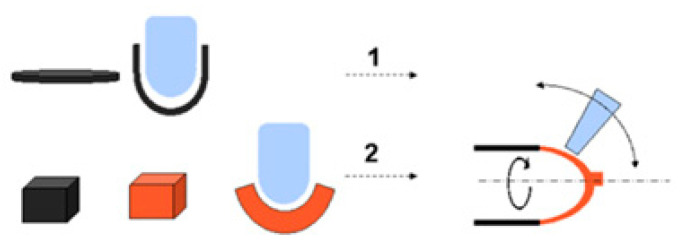
Manufacturing process of the steel vessel. (1) Halves manufacturing by cold drawing. (2) End caps manufactured by repulse technic.

**Figure 2 sensors-23-03134-f002:**
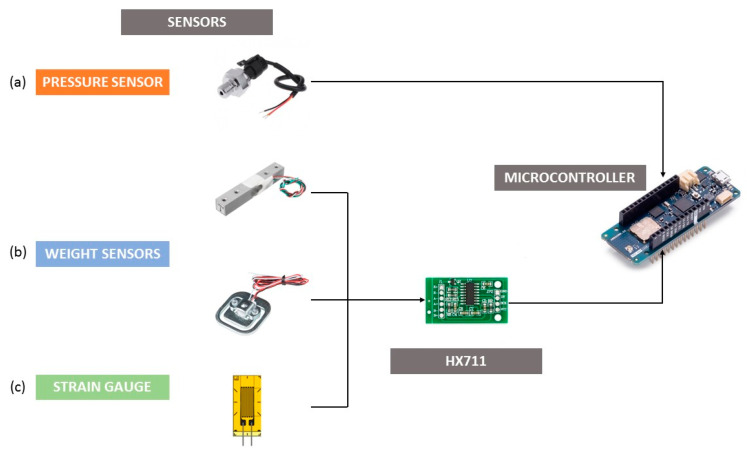
Sensors and controller used.

**Figure 3 sensors-23-03134-f003:**
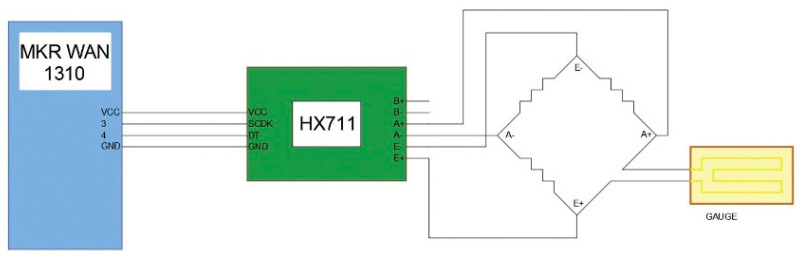
Wheatstone bridge prototype.

**Figure 4 sensors-23-03134-f004:**
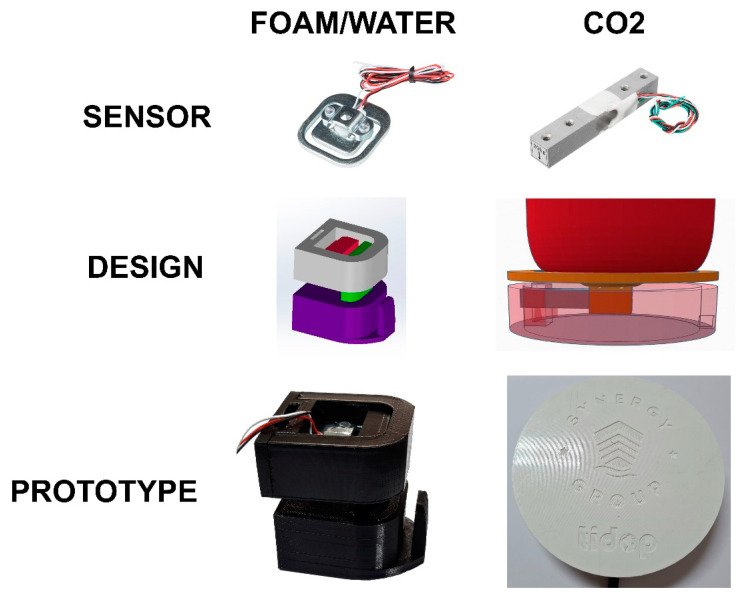
Sensors, designs and prototypes for the measurement of weight for different types of extinguishers (powder/water and CO_2_).

**Figure 5 sensors-23-03134-f005:**
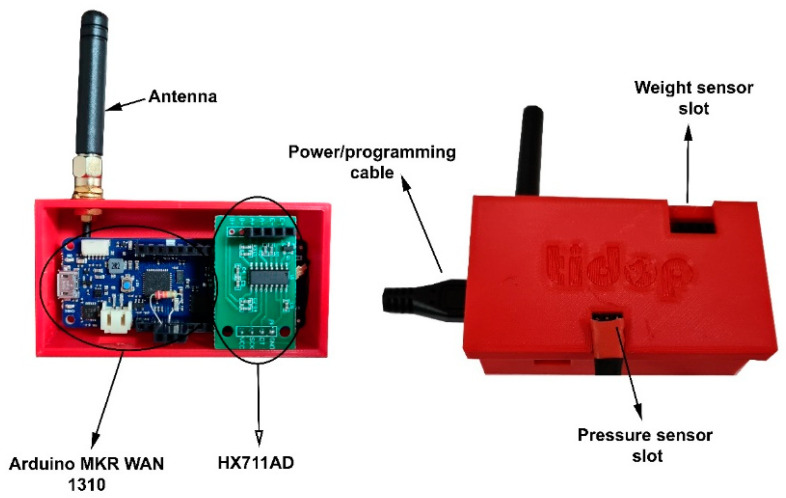
**Data acquisition system and different slot designs.** Sensors, designs and prototypes for the measurement of weight for different types of extinguishers (powder/water and CO_2_).

**Figure 6 sensors-23-03134-f006:**
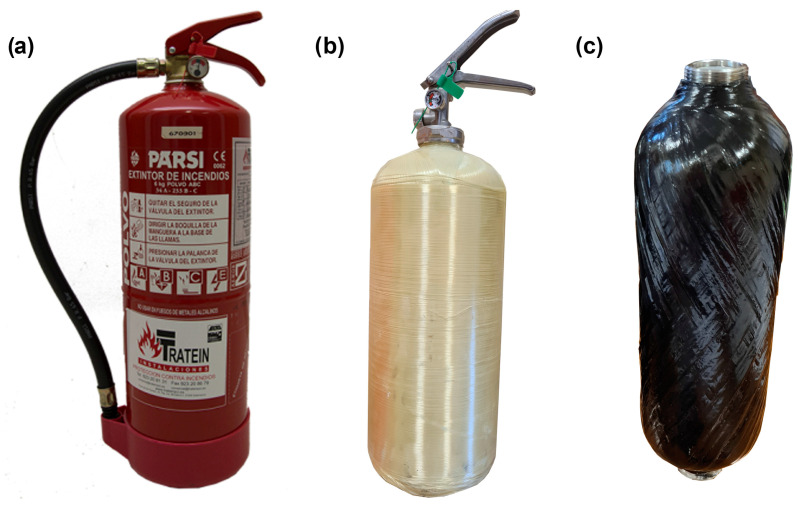
Vessels studied. (**a**) Conventional steel-based vessel; (**b**) AFRP-based vessel; and (**c**) CFRP-based vessel.

**Figure 7 sensors-23-03134-f007:**
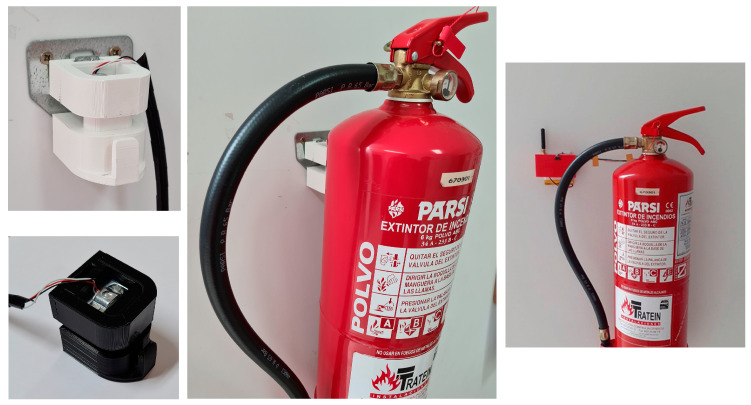
Powder hook design and complete system assembled.

**Figure 8 sensors-23-03134-f008:**
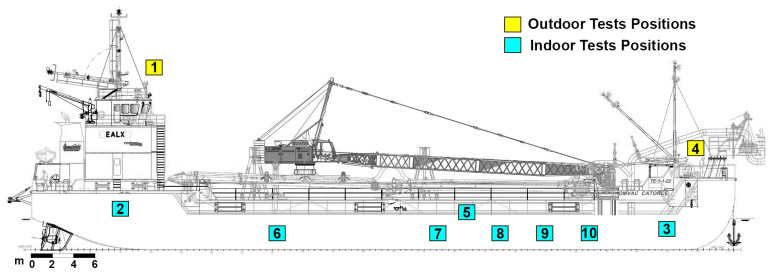
Antennas and coverage test placements within the ship.

**Figure 9 sensors-23-03134-f009:**
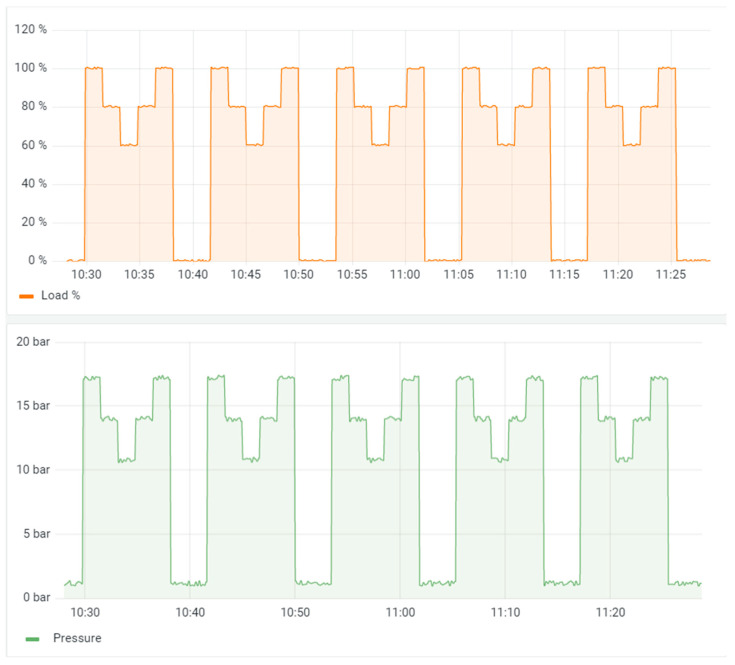
Dashboard through a measurement test session. The graph above represents the percentage of load with respect to the setting. The bottom graph represents the actual pressure reading of the device.

**Table 1 sensors-23-03134-t001:** Types of COPV.

Type	Liner Material	Fiber Material	Kg/L
Type I	Aluminum	Aluminum	1.40
Type II	Aluminum	Glass fiber	0.98
Type III	Aluminum	Carbon fiber	0.74
Type IV	Polymer	Carbon/hybrid fiber	0.45
Type V	-	Carbon/hybrid fiber	0.32

**Table 2 sensors-23-03134-t002:** Mechanical properties of the fibers.

	Density	Young’s Modulus (GPa)	Tensile Strength (MPa)	Elongation at Maximum Tensile Strength
**Steel alloy**	7.86	210	4830	38.0
**Carbon fiber**	1.76	230–260	≥4000	1.5
**Glass fiber**	2.44	100–230	≥3400	4.5
**Aramid fiber**	1.44	125	≥3400	2.4

**Table 3 sensors-23-03134-t003:** Principal properties of the materials employed to make pressure vessels.

	Low-Pressure			High-Pressure		
	Total Weight	Maximum Admissible Pressure (bar)	Price EUR/kg	Total Weight	Maximum Admissible Pressure (bar)	Price EUR/kg
**Steel alloy**	9.00	70	1.80	14.59	165.00	1.80
**Carbon fiber**	6.67	80	33.15	7.15	188.57	33.15
**Glass fiber**	6.93	68	16.23	7.98	160.29	16.23
**Aramid fiber**	0.55	68	15.73	6.76	160.29	15.73

**Table 4 sensors-23-03134-t004:** Steel Vessel Characteristics.

Height (mm)	525
Diameter (mm)	150
Thickness (mm)	1.5
Volume (L)	6.70
Empty Weight (kg)	3.20
Total Weight (kg)	9.28

**Table 5 sensors-23-03134-t005:** AFRP vessel characteristics.

Height (mm)	420
Diameter (mm)	170
Thickness (mm)	1
Volume (L)	6.5
Empty Weight (kg)	0.842
Total Weight (kg)	6.842

**Table 6 sensors-23-03134-t006:** CFRP vessel characteristics.

Height (mm)	310
Diameter (mm)	100
Thickness (mm)	2
Volume (L)	6
Empty Weight (kg)	0.503
Total Weight (kg)	6.503

**Table 7 sensors-23-03134-t007:** Comparison between different vessels.

	Steel	Aramid	Carbon
Height (mm)	525	420	310
Diameter (mm)	150	170	100
Thickness (mm)	1.5	1	2
Volume (L)	6.70	6.50	2
Empty Weight (kg)	3.20	0.842	0.503
Total Weight (kg)	9.28	6.842	6.503
Pressure (bar)	70	68	80

**Table 8 sensors-23-03134-t008:** Coverture test results for SF 7.

		Gate Indoor		Gate Outdoor	
Point	Airtime (s)	RSS in (dBm)	SNR in (dB)	RSSI Out (dBm)	SNR Out (dB)
1	0.051456	−31	9.25	−64	9.25
1	0.051456	−27	9.25	−73	9.25
2	0.051456	−37	7.5	−47	7.5
2	0.051456	−75	7	−92	7
3	0.051456	−79	7.25	−89	9
3	0.051456	−83	7.25	−73	9
3	0.051456	−83	4	−86	7.25
4	0.051456	−84	3.5	−77	10.5
4	0.051456	−75	6.75	−79	8
4	0.051456	−73	7.5	−75	8.25
5	0.051456	−75	6.25	−65	6.5
5	0.051456	−85	3.25	−78	8.75
5	0.051456	−87	0.5	−77	7.25
6	0.051456	−85	5	−81	6.75
6	0.051456	−89	1	−83	7.25
7	0.051456	−85	2.75	−90	8.75
7	0.051456	−91	−5.25	−95	9.5
8*	0.051456	**	**	−97	8.75
8*	0.051456	**	**	−99	5.25
8*	0.051456	−87	−7.75	−93	9.25
9*	0.051456	**	**	−93	6
9*	0.051456	**	**	−95	8.5
10*	0.051456	−96	−6.5	−93	7.5
10*	0.051456	**	**	−93	6.75
10*	0.051456	**	**	−91	9
10*	0.051456	**	**	−97	5.5

* SNR above limit (7.5 dB). ** Lost packages.

**Table 9 sensors-23-03134-t009:** Coverture test results for SF 9.

		Gate Indoor		Gate Outdoor	
Point	Airtime (s)	RSS in (dBm)	SNR in (dB)	RSSI Out (dBm)	SNR Out (dB)
1	0.185344	9	−37	9	−65
2	0.185344	9	−72	9	−83
2	0.185344	9	−71	11.25	−85
2	0.185344	9	−77	10.5	−87
3	0.185344	9	−86	0.75	−79
3	0.185344	9	−85	1.25	−73
3	0.185344	9	−92	2.5	−80
4	0.185344	9	−74	11	−75
4	0.185344	9	−85	4.25	−67
4	0.185344	9	−73	9.25	−77
5	0.185344	9	−87	6	−91
5	0.185344	9	−86	6.75	−80
5	0.185344	9	−83	7.5	−78
6	0.185344	9	−90	−3	−79
6	0.185344	9	−90	−1.5	−87
6	0.185344	9	−91	5	−86
7	0.185344	9	−92	−5.75	−90
8*	0.185344	9	−88	−9.75	−96
8*	0.185344	9	−90	−6.5	−104
8*	0.185344	9	**	**	−95
9*	0.185344	9	**	**	−97
9*	0.185344	9	**	**	−109
10*	0.185344	9	−89	−4.75	−99
10*	0.185344	9	**	**	−101
10*	0.185344	9	−89	−11	−101
10*	0.185344	9	**	**	−107

* SNR above limit (7.5 dB). ** Lost packages.

**Table 10 sensors-23-03134-t010:** Coverture test results for SF 11.

		Gate Indoor		Gate Outdoor	
Point	Airtime (s)	RSS in (dBm)	SNR in (dB)	RSSI Out (dBm)	SNR Out (dB)
1	0.659456	−13	10.75	−61	11.5
2	0.659456	−74	8.5	−86	10.5
3	0.659456	−83	2075	−73	11.25
3	0.659456	−74	6	−75	9025
3	0.659456	−74	805	−70	11
4	0.659456	−72	8025	−62	9
5	0.659456	−89	1	−78	10
5	0.659456	−87	−0025	−76	9.5
6	0.659456	−85	2075	−78	10.25
6	0.659456	−85	6.75	−104	5075
7	0.659456	−93	−7.5	−91	7
8*	0.659456	**	**	−89	7.25
9*	0.659456	**	**	−103	−0.25
9*	0.659456	**	**	−97	4.75
10	0.659456	−90	−9.5	−105	1.25

* SNR above limit (7.5 dB). ** Lost packages.

**Table 11 sensors-23-03134-t011:** Statistic study of the values.

Real Pressure	Value	Mean	Standard Deviation	Variation Coefficient	Mean Noise
17 bar	Load (%)	1.0041	0.0049	0.492	0.0025
	Pressure (bar)	17.1845	0.1216	0.708	0.0642
14 bar	Load (%)	0.8049	0.0031	0.382	0.0018
	Pressure (bar)	13.9940	0.1313	0.939	0.0724
11 bar	Load (%)	0.6055	0.0030	0.487	0.0017
	Pressure (bar)	10.8140	0.1172	1.084	0.0657
1 bar	Load (%)	0.0048	0.0029	0.602	0.0017
	Pressure (bar)	1.1933	0.1164	0.098	0.0696

## Data Availability

Data sharing not applicable.

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
