# Peer review of "Smartvessel: A New Extinguisher Prototype Based on New Materials and IoT Sensors"

_sensors, 2023, doi:10.3390/s23063134_

Round 1
Reviewer 1 Report
The topic of the work is very interesting, useful, and popular.
The literature in the introductory part is adequate and sufficient, however, it is also used a lot in the rest of the paper, where the own results should dominate. Additionally, details of the work are missing and the results are poorly explained. For example, it was said that a piece is made of Polyethylene Terephalate Glycol (PETG) and printed in 3D. A little more detail about that piece should be given as the type of additive 3D technology is being used.
In the text of the paper, the authors refer first to figure 3 before figure 2. In addition, the figures do not contain the parts a, b, and c that are referred to in the text.
It could be nice if the authors also give a final picture of the entire system with the integrated sensors in the housing connected to the electronic system.
For CFRP Vessel it is emphasized in the text that a total load is 6 kg, while from table 6 it can be concluded that the total load is 2 kg.
Author Response
The authors would like to thank the reviewers for their efforts and their useful and constructive comments. The manuscript has been thoroughly revised according to their indications. New contributions in the manuscript have been highlighted.
All answers can be found in the attached document.

Reviewer 2 Report
In this paper, the authors are focused on the evaluation of extinguishers that integrate monitoring sensors and intelligent maintenance.
As general remark, it is unclear what is new in the paper: the manufacturing process of the extinguisher or the IoT solution (sensors, accessories and connectivity).
From my point of view there are some aspects to improve:
1. Abstract does not contain any quantitative data about the results. The abstract should be revised and improved.
2. General properties of known materials are presented in the Table 2, but specific properties of materials used are missing. What types of carbon, glass or kevlar fibers were used? What provider?
3. The parameters of the Filament Winding manufacturing process are missing. The manufacturing process of composite material extinguisher should be detailed specified, pointed on what is new.
4. How many layers of composite materials were used to obtained as example a 1 or 2 mm thickness wall?
5. The composite materials used should be clearly described. What type of epoxy resin (provider) was used?
6. Some components were 3D printed. Please mention the type of the 3D printer.
7. Figure 8 is unclear. The distances between different positions should be specified, using a reference system.
8. "To monitor and graphically represent the data in real time, a dashboard was developed, showing the different measurements through time". What programming language was used for dashboard design?
9. The dashboard should be used to check the repeatability of the experiments. Usually, 3 or 5 experiments should be done in the same conditions and the resulting data should be statistically analyzed.
10. The conclusions should be improved based on the results.
11. Are the limitations of this study noted? The limitations of this study should be discussed.
Author Response

(The authors gave the same response as above.)

Round 2
Reviewer 1 Report
The authors took into consideration all the comments.
The revised version is improved.
Reviewer 2 Report
All the comments are addressed well and utilized to improve the manuscript. The manuscript is acceptable.